# Biofilm Formation and Antibiotic Resistance Profiles in Carbapenemase-Producing Gram-Negative Rods—A Comparative Analysis between Screening and Pathological Isolates

**DOI:** 10.3390/antibiotics13080687

**Published:** 2024-07-24

**Authors:** Camelia Vintilă, Răzvan Lucian Coșeriu, Anca Delia Mare, Cristina Nicoleta Ciurea, Radu Ovidiu Togănel, Anastasia Simion, Anca Cighir, Adrian Man

**Affiliations:** 1Department of Microbiology, George Emil Palade University of Medicine, Pharmacy, Science and Technology of Târgu Mureș, 540142 Târgu Mures, Romania; camelia.vintila@umfst.ro (C.V.); anca.mare@umfst.ro (A.D.M.); cristina.ciurea@umfst.ro (C.N.C.); toganel.radu-ovidiu.22@stud.umfst.ro (R.O.T.); anastasia.simion@umfst.ro (A.S.); anca.cighir@umfst.ro (A.C.); adrian.man@umfst.ro (A.M.); 2Doctoral School of Medicine and Pharmacy, George Emil Palade University of Medicine, Pharmacy, Science and Technology of Târgu Mures, 540142 Târgu Mures, Romania

**Keywords:** biofilms, screening isolates, clinical isolates, carbapenemase, resistance genes

## Abstract

(1) Background: Carbapenem-resistant (CR) bacteria pose a significant global public health challenge due to their ability to evade treatment with beta-lactam antibiotics, including carbapenems. This study investigates the biofilm-forming capabilities of CR clinical bacterial isolates and examines the impact of serum on biofilm formation. Additionally, the study evaluates the resistance profiles and genetic markers for carbapenemase production. (2) Methods: Bacterial isolates were collected from the microbiology laboratory of Mures County Clinical Hospital between October 2022 and September 2023. Pharyngeal and rectal swabs were screened for carbapenem-resistant bacteria using selective media. Lower respiratory tract samples were also analyzed for CR Gram-negative bacteria. The isolates were tested for their ability to form biofilms in the presence and absence of fetal bovine serum at 24 and 48 h. Carbapenemase production was detected phenotypically and confirmed via PCR for relevant genes. (3) Results: Out of 846 screened samples, 4.25% from pharyngeal swabs and 6.38% from rectal swabs tested positive for CR bacteria. *Acinetobacter baumannii* and *Klebsiella pneumoniae* were the most common species isolated. Biofilm formation varied significantly between clinical isolates and standard strains, with clinical isolates generally showing higher biofilm production. The presence of serum had no significant effect on biofilm formation in *Klebsiella* spp., but stimulated biofilm formation for *Acinetobacter* spp. Carbapenemase genes *bla*_KPC_, *bla*_OXA-48-like_, and *bla*_NDM_ were detected in various isolates, predominantly in *Klebsiella* spp., but were not the main determinants of carbapenem resistance, at least in screening isolates. (4) Conclusions: This study highlights the variability in biofilm formation among CR clinical isolates and underscores the differences between the bacteria found as carriage versus infection. Both bacterial species and environmental factors variably influence biofilm formation. These insights are crucial for the development of effective treatment and infection control strategies in clinical settings.

## 1. Introduction

Carbapenem-resistant bacteria (CR) pose a global public health threat, often limiting the treatment options for the infections that they cause. In recent years, statistics indicate almost 5 million deaths associated with antimicrobial-resistant bacteria [1]. Carbapenem-resistant bacteria have evolved to be able to survive in the presence of therapeutic doses of beta-lactam antibiotics, including carbapenems. This resistance can result from carbapenemase production, as seen in carbapenemase-producing *Enterobacteriaceae* (CPE), or from other mechanisms, such as efflux pumps or impermeability, mostly in non-fermentative Gram-negative bacilli.

The European Center of Disease Control (ECDC) has published recommendations for infection prevention and control, through screening programs, for the early detection of carriers and outbreak management. These measures include CPE rectal screening at hospital admission or before admission to a specific ward, such as an Intensive Care Unit (ICU). Moreover, other CR bacteria are equally essential to be screened. The presence of Gram-negative bacteria in the pharynx or intestine is significant, as demonstrated by prior research. CPE screening can detect the colonization and the potential for biofilm formation, a risk that is particularly elevated in mechanically ventilated patients in the pulmonology department or ICU [2,3]. Understanding the behavior of these bacteria in biofilms is crucial for the development of effective treatment strategies.

Biofilms are complex communities of bacteria typically attached to a surface. They differ physiologically from commensal (or planktonic) bacteria, which have the potential to transition into biofilm-forming bacteria under certain conditions [4]. Planktonic bacteria grow faster, with higher virulence, but have increased susceptibility to antibiotics compared to biofilm-related phenotypes. Within a mucous membrane biofilm, bacteria can develop resistance to host immunity and also to antibiotics [5,6]. Bacterial growth can be influenced by a variety of factors that are part of the inner body, in terms of either the potential stimulation (albumin, growth factors, amino acids, sugars, lipids) or the inhibition (immunoglobulins, complement fractions, transferrin) of bacterial growth [7,8,9,10]. Thus, fetal bovine serum is an in vitro alternative to simulate the effects of serum activity on biofilm formation [11,12,13].

Both Gram-negative and Gram-positive bacteria are capable of developing biofilms, especially when they are associated with one another [6,14,15]. The spreading of multi-drug resistant bacteria in a medical unit, especially from one patient to another, is the worst-case scenario feared by every clinician. Distinguishing between CPE and non-CPE bacteria holds significance in infection control and epidemiological contexts due to the possibility of passing on mobile genetic elements, including resistance genes. Despite this, the identification of carbapenem resistance mechanisms is not currently advised for treatment decisions and is not commonly performed in most clinical laboratories. On the other hand, differentiation between CPE and non-CPE remains crucial for infection control and epidemiological surveillance due to the widespread occurrence of carbapenemases [16].

The current study aimed to evaluate the biofilm-forming ability of clinical CR bacterial isolates, considering their origin (whether from a carrier state or from infections) and the influence of serum as a biological factor. Additionally, the evaluation of the resistance profile and the performance of a genetic analysis for the presence of carbapenemase genes enhance our understanding of the CR resistance mechanisms. This knowledge is essential for the development of improved patient treatment strategies and to facilitate better diagnostic management.

## 2. Results

Screening was concluded in a total of 846 samples that were processed for the detection of CR Gram-negative bacteria in one year (October 2022–September 2023), of which 4.25% (*n* = 36) were found positive with one or more species from a throat swab and 6.38% (*n* = 54) from a rectal swab. Of these, 26 strains from pharyngeal swabs fulfilled the inclusion criteria, being isolated from patients with an average age of 47.88 years old (SD = 21.94), predominantly males (*n* = 22), with 46.15% originating from an urban environment. The most encountered bacterial species was *Acinetobacter baumannii*, representing 42.30% (*n* = 11), followed by *Stenotrophomonas maltophilia* with 23.07% (*n* = 6) and *Klebsiella pneumoniae* with 19.23% (*n* = 5). From the total number of patients admitted to the ICU and who were found positive upon CR bacterial screening, 76.16% (*n* = 16) died. In seven patients, the screening revealed similarities regarding the isolated species from rectal and pharyngeal swabs. Following the antibiogram, 19 samples isolated were considered PDR (90.47%), while the rest of the strains were classified as XDR. The cumulative antibiogram is presented in Table 1.

The analysis of bacterial strains isolated from pathological samples (*n* = 30) showed that *K. pneumoniae* and *A. baumannii* were the most frequently encountered species in the lower respiratory tract (*n* = 13), followed by *Pseudomonas aeruginosa* in four cases. The isolates were derived from patients with an average age of 74 years old (SD = 15.42), with a predominance of the male sex (*n* = 25, 83.33%).

The patients were initially admitted to various wards, as presented in Figure 1, but were subsequently transferred to the ICU due to their deteriorating health conditions.

### 2.1. Biofilm Results

The evaluation of biofilm formation by the *Klebsiella pneumoniae* strains isolated from the screening samples showed that the average OD was 0.11 at 24 h incubation, compared with the clinical isolates, with OD = 0.834. Bovine serum did not affect the amount of biofilm produced by *Klebsiella* spp. (ANOVA test, CI 95%, *p* = 0.247), as can be seen in Figure 2, irrespective of the bacterial origin. Additionally, the incubation time had no significant effect on biofilm formation, with measurements taken at 24 and 48 h showing no significant differences (*p* = 0.08), except in strains 1 and 3 in the presence of bovine serum and 2 and 3 without serum, with an improvement in biofilm formation at 48 h, as presented in the image below. At 24 h, there were no significant differences in biofilm formation between the control strain and the CR *Klebsiella* strains isolated from screening. However, the carbapenemase-producing *Klebsiella* control strain (ATCC BAA1705) showed more efficient biofilm formation, both with and without bovine serum, at both time points. At the 24 h incubation time, the presence of bovine serum inhibited the formation of biofilms in two out of five strains, comparable to that at 48 h incubation.

One of the 13 tested clinical isolates (7.69%) exhibited significantly higher biofilm formation compared to the others, as shown in Appendix A and Figure 3, at both 24 and 48 h. Similarly, the *K. pneumoniae* ATCC BAA1705 strain demonstrated robust biofilm production, inhibited by bovine serum at 24 h but conversely stimulated at 48 h. Regarding the influence of the incubation time on biofilm formation in the presence of bovine serum, three out of the 13 strains (23.07%) showed lower biofilm formation at 48 h compared to 24 h (*p*-value < 0.05, CI 95%). Moreover, at 24 h of incubation, a significant difference was observed in biofilm formation among four out of the 13 strains (30.76%) (*p*-value < 0.05, CI 95%), with a notable decrease in absorbance from 4.9 to 2.03.

In the case of *Pseudomonas aeruginosa* screening isolates (Figure 4), significant differences were observed in the biofilm formation capacity compared to other species, both in the presence and absence of bovine serum, at 24 and 48 h, with a few exceptions. The ANOVA tests revealed a significant difference between *Pseudomonas* screening strains (*p*-value < 0.0001, 95% confidence interval) in biofilm formation in the presence of bovine serum at 24 h. Post hoc Tukey–Kramer tests identified differences between the strains, with similar results at 48 h.

For the clinical isolates of *Pseudomonas* (Figure 5), two out of four strains (50%) exhibited significantly superior biofilm formation at 24 h in the absence of bovine serum. However, there was a slight increase in biofilm formation at 48 h, with significant differences noted between the two strains, suggesting the potential overactivation of enzyme production involved in biofilm formation. The presence of bovine serum positively influenced biofilm formation at both 24 and 48 h. The statistical analysis revealed significant differences between the strains (*p*-value < 0.05), confirming the impact of bovine serum on biofilm formation.

The isolates of *Acinetobacter* from screening demonstrated a wide variety of biofilm formation capacities, with two strains showing the most significant increase in activity. Significant statistical differences were observed between the strains when comparing incubation at 24 and 48 h, in the presence and absence of bovine serum. Furthermore, differences were noted between the strains at 24 h of incubation, and even more so at 48 h, with a negative impact on biofilm formation.

Among the clinical isolates of *Acinetobacter*, two strains exhibited increased biofilm activity, mainly in the presence of serum and to a lesser extent in its absence (*p* < 0.05), both at 24 and 48 h of incubation, yielding absorbance values between 1.1 and 3.3, as presented in Figure 6.

*Stenotrophomonas* spp. was isolated only from screening, and notable variations were observed in biofilm formation in the presence and absence of bovine serum among certain strains, particularly noticeable at the 48 h mark, where most strains did not exhibit enhanced biofilm formation compared to the standard ATCC strain (Figure 7). *Stenotrophomonas* presented better biofilm activity in the absence of bovine serum in four of the six strains (66.67%), demonstrating inhibitory serum activity also from this bacterial species.

As presented above, seven strains were isolated both from pharyngeal swabs and from rectal swabs, namely *K. pneumoniae*, *Pseudomonas aeruginosa*, *S. maltophilia* (*n* = 2 for each), and *A. baumannii* (*n* = 1). When comparing each pair in the absence of bovine serum, the Bonferroni post hoc statistical test revealed differences only for *A. baumannii*, with the rectal isolates exhibiting increased biofilm production (*p*-value < 0.001, CI 95%). In the presence of bovine serum, all *A. baumannii* isolates demonstrated enhanced biofilm formation at 24 h, with a significantly greater increase at 48 h.

### 2.2. Genetic Test Results

All samples analyzed for biofilm formation (*n* = 56) were also tested for their ability to produce carbapenemases, both genetically and phenotypically. The most encountered carbapenemase gene was *bla*_OXA-48-like_ (*n* = 12; 24%), and, of these, four strains also presented *bla*_NDM_ (26.66%). One strain of *K. pneumoniae* presented *bla*_VIM_. The results for the other analyzed genes (*bla*_SPM_, *bla*_IMP_, and *mcr*-1) were negative. As presented in Table 2, most genetic carbapenemase determinants were found in *K. pneumoniae*, with *bla*_OXA-48-like_ being the most predominant (*n* = 10), followed by *bla*_NDM_ (*n* = 4) and *bla*_VIM_ (*n* = 1). There were 11 strains with genes that produced biofilms with an OD above 0.1, with no association between the presence of carbapenemase genes and the ability of the bacteria to produce biofilms (OR = 1.4, *p* = 1.0). Only one strain of *P. aeruginosa* and one of *A. baumannii* presented *bla*_OXA-48-like_. The results of the immunochromatography assay for the phenotypic detection of carbapenemases were consistent with the genetic findings, namely *bla*_NDM_ (*n* = 4), *bla*_OXA-48-like_ (*n* = 11) and no *bla*_KPC_.

## 3. Discussion

The carriage of MDR bacteria constitutes a major concern, especially for patients admitted to critical departments, such as ICUs or surgical wards. Possible colonization with these types of bacteria represents a source of cross-transmission for multiple resistance genes, such as extended-spectrum β-lactamases, carbapenemases, or even mediated colistin resistance genes [17]. Nevertheless, the reality is that this phenomenon is omnipresent, as presented in many publications. For example, a prospective study conducted by Schoevaerdts et al. in a geriatric ward in Belgium revealed the presence of *bla*_CTX-M_, *bla*_TEM-52-like_, *bla*_SHV-2-like_, and other genes in 39 newly admitted patients [18].

The screening of colonization with MDR bacteria is important not only for hospital admissions but also for other institutes, such as long-term care facilities, as was demonstrated in a study conducted by Le et al. in Japan. The study demonstrated the presence of MDR *Acinetobacter* spp., *Enterobacterales*, and *Pseudomonas* in 38% of the tested patients, using the same methodology as in our study [19]. Another study, conducted in Germany in 2013, detected methicillin-resistant *S. aureus* MRSA, extended-spectrum β-lactamase-producing enterobacteria, and even vancomycin-resistant enterococci from the nose, throat, and perineum [20]. Similarly, in 2014, 2.5% MDR *P. aeruginosa* was detected from 641 pharyngeal or stool swabs [21]. In many cases, the presence of MDR bacteria may not be associated with the primary disease, with the patients being considered “carriers” [22]. In all of the presented studies, the average age of the patients was 64–68 years, indicating that the presence of this type of bacteria is more frequent in the elderly, who are more vulnerable to other types of infection. In our study, the average age was even higher (74 years old), mostly admitted to the ICU ward. In some other countries, such as India, the rate of the fecal carriage of MDR bacteria is between 18 and 73% in ICU wards, showing the importance of screening methods in order to detect these types of bacteria. Of 140 rectal swabs, 23.57% were CR *Escherichia coli*, followed by 10.71% *Klebsiella* species [23]. Still, an important question remains: does colonization with MDR bacteria contribute to biofilm formation?

It is well known that bacteria are able to grow and adhere to different surfaces, such as plastic, metal, and others, but can also develop in vivo biofilms on prosthetic devices and mucosal membranes. The identification of different strains from an endotracheal tube (ETT), for example, is not so rare. Mechanically ventilated patients present, in the first 7 days, a 90% chance of developing a bacterial biofilm on the ETT, as the most common cleaning procedures do not provide a protective effect against this [24]. As shown in the results above, only one strain of *Klebsiella* exhibited significant biofilm formation (7.96%). This contrasts with other studies, where 64–85% of *Klebsiella* isolates produced intense or moderate biofilms [25,26].

Pathogens can grow and replicate even under unfavorable conditions, as shown for *Acinetobacter* strains. Different studies have shown that the ability of *Acinetobacter* to form biofilms is highly variable, from 11% to 74% or even 90% when the material of the attachment substrate was changed [27,28]. It seems that this is a particular trait for *Acinetobacter* spp., as presented in other research, where this species was able to develop a mature biofilm when grown in human serum [29].

In our results, *A. baumannii* demonstrated the highest biofilm-forming capacity among all tested species, particularly in strains isolated from patients’ pathological samples compared to screening isolates. This was potentially due to the presence of biofilm-associated proteins such as *bap* or due to the modification of the outer membrane protein *OmpA*. Another factor could be the presence of pili, which facilitate attachment and promote biofilm formation. In *A. baumannii*, these pili cluster together in the form of an operon with a pilus-like bundle structure, further enhancing the bacterial adaptability in the form of biofilms [30]. The transition from a carriage state to an infection state is often influenced by host factors (such as the immune status), environmental factors (such as antibiotic use), and bacterial factors (such as the expression of virulence genes). During infection, bacteria may express virulence factors that were previously dormant in the state of carriage or change their behavior following the stress related to antibiotic therapy [31,32].

As for the implications of bovine serum in the development of biofilms, our results showed its inhibitory activity against most of the tested strains. This can be explained by the presence of variate factors such as transferrin, as described mostly for *P. aeruginosa* [33]. Antimicrobial peptides from the serum, such as immunoglobulins, complement proteins, proteases, or lactoferrin, as well as nutrient or ion chelation, are also involved in biofilm reduction. Moreover, inhibitory metabolites may impair bacterial quorum sensing, disrupting their ability to organize into biofilms [34,35,36]. Individual bacterial traits lead to the uniqueness of the pathogenic strain. For example, it has been found that *P. aeruginosa* strains that are unable to produce pyoverdine will develop a much thinner biofilm due to the inability to acquire ions [37], and serum ion chelators may further impair iron acquisition and thus biofilm formation.

Our initial hypothesis involved the possible association of carbapenemases and biofilm formation, but no correlations. Other authors have studied the possible relation between antimicrobial resistance and the capacity for biofilm formation in multiple types of strains, but no clear correlation was found either [26]. Although it was shown that bacteria producing NDM-4 carbapenemases downregulate the biofilm activity, under meropenem stress, they will produce more flagellar, fimbria, and pilus proteins, contributing to increased biofilm formation [38]. Other factors can influence biofilm formation, divided into biochemical, molecular, or altered host factors, all contributing to the diversity of bacteria [39]. We have also demonstrated that the diversity of bacteria is a reality, in terms of the heterogeneous behavior of the individuals that are part of a certain species.

*Klebsiella* species presented a large number of carbapenemase genes, comparable with other studies conducted in Turkey, where 58% of the *Klebsiella* strains presented *bla*_OXA-48-like_, but only 2% presented *bla*_NDM_ [40]. When comparing the strains isolated from screening with clinical isolates, a higher prevalence of carbapenemase genes was noted in the latter group. This raises an important question: why did the initial screening for carbapenem-resistant bacteria at the time of ICU admission detect strains without carbapenemase genes, yet, a few days later, carbapenemase-producing bacteria of the same species were isolated from the lower respiratory tract? It is possible that several factors, such as the therapeutic practices, the initial hospital department of admission, and disinfection and sterilization methods, contribute to the transmission of resistance genes among bacteria or the selection and spread of carbapenemase-producing strains. The management of infections associated with the medical act should be carefully and thoroughly implemented and followed.

## 4. Materials and Methods

The study was conducted on bacterial isolates collected from the microbiology laboratory of Mures County Clinical Hospital (MCCH) between October 2022 and September 2023. Retrospective information about the patients was collected from the data available in reports, including, but not limited to, the age, sex, admission period, and previous antibiotic or antifungal treatments. The approval of the Ethical Board of MCCH (No. 2618 from 12 April 2024) was obtained.

### 4.1. Sample Selection

The samples were processed in the microbiology laboratory, complying with the internal protocols, as follows: to screen for the presence of CR bacteria, pharyngeal and rectal swabs were taken from the admitted patients, which were then inoculated on CRE Brilliance Agar (Oxoid, Hampshire, UK) selective media and incubated at 35 °C for 16–18 h. The colonies were identified to species using classical phenotypic methods. The bacteria were identified based on their morphology (Gram staining), culture, and biochemical nature (sugar fermentation; protein metabolism; enzyme production—catalase, oxidase; hemolysis, pigment production). When needed, identification was performed using automated methods (Vitek 2 Compact, Biomerieux, Marcy-l’Étoile, France). Meropenem resistance was confirmed by disk diffusion for all isolates, using 10 µg disks.

Lower respiratory tract clinical samples (tracheal aspirate, bronchial aspirate, intra-oro-tracheal tube, and sputum) were collected from ICU-admitted patients. These samples were inoculated on Columbia agar and CLED agar (cystine lactose electrolyte-deficient) and incubated at 35 °C for 16–18 h. The Gram-negative bacteria colonies were identified using standard methods (based on biochemical and phenotypic characteristics) or through automatic identification using the Vitek 2 Compact (Biomerieux, Marcy-l’Étoile, France), and they were stored at −70 °C for further experiments.

As the study focused on the microbiological aspects of respiratory tract infections, the inclusion criteria for the isolates to be included in the study were as follows:–all isolates were grown on CRE Brilliance Agar (Oxoid, Hampshire, UK), from the upper respiratory tract (pharyngeal swab), and confirmed to present a carbapenem-resistance phenotype, identified to the genus and species;–they were multi-drug-resistant (MDR), extensively resistant (XDR), or pan-resistant (PDR) Gram-negative bacterial isolates from the ICU ward, from the lower respiratory tract.

All isolates were tested for antimicrobial susceptibility using the Kirby–Bauer diffusion method and MIC. The interpretation was performed according to the European Committee on Antimicrobial Susceptibility Testing (EUCAST) and the result for each isolate was reported as susceptible, intermediate (susceptible at a higher dose), or resistant. An extended antibiogram was performed for the clinical isolates.

### 4.2. Evaluation of Ability of Bacteria to Form Biofilms

As was previously described in other studies, the formation of biofilms might be influenced by a variety of factors, such as the substances, environmental conditions, or time. For this, the study evaluated biofilm formation in the presence and the absence of fetal bovine serum and at different incubation times (24 h and 48 h) for all strains identified in clinical pathological products or screening [3,41,42].

### 4.3. Biofilm Culture

All bacterial isolates were cultured on CLED agar, to obtain a fresh culture. To evaluate biofilm formation, two separate experiments were conducted. For each experiment, 190 µL of RPMI-1640 medium (Sigma-Aldrich, St. Louis, MO, USA) was added to the wells of 96 microtiter plates, in sterile environmental conditions. A 0.5 McFarland bacterial inoculum was prepared in saline solution, and, from this, 10 µL was added in 3 corresponding wells (for triplicate reading), and it was thoroughly mixed. In the second plate, 20% fetal serum bovine was supplementarily added to the wells. The final volume per well was 200 µL. Aside from the clinical isolates, standard ATCC strains were used as follows: *K. pneumoniae* ATCC 13883, *K. pneumoniae* ATCC BAA 2470, *P. aeruginosa* ATCC 27853, *S. maltophilia* ATCC 17666. A negative control consisting of a mixture of 190 µL culture medium and 10 µL saline solution without bacteria was also used to prove the sterility of the reagents.

### 4.4. Biofilm Growth Assay

Following incubation, the RPMI medium was discarded by carefully inverting the plate. Non-adherent or loosely attached bacteria were carefully removed by immersing the plate in sterile distilled water. This process was repeated three times.

To assess biofilm formation, 200 µL of crystal violet 0.1% (aqueous solution) was added to each well and they were incubated at room temperature for 10 min. During this step, the bacteria attached to the plastic walls in the form of biofilms were stained. To discard the excess crystal violet, the plates were washed with sterile distilled water three times.

### 4.5. Acetic Acid Treatment and Spectrophotometry Analysis

The disintegration of the bacterial cells and disruption of the biofilm matrix were performed by adding 200 µL of 30% acetic acid to the wells and incubating them at room temperature for 10 min. The intensity of the color was read by spectrophotometry in the ThunderBolt Virclia (Vircell, Granada, Spain) analyzer at a wavelength of 560 nm.

### 4.6. Evaluation of Ability of Bacteria to Produce Carbapenemases

The phenotypic detection of carbapenemases was performed with the RESIST-3 O.K.N K-SeT (Coris Bioconcept, Gemblousx, Belgium) immunochromatography assay, which can detect 3 important carbapenemases: KPC, OXA, and NDM. The detection of the corresponding genes (*bla*_KPC_, *bla*_OXA-48-like_, and *bla*_NDM_), but also other relevant carbapenemase genes (*bla*_GES-2_, *bla*_IMP_, *bla*_VIM_, *bla*_SPM_), was performed by endpoint PCR. The correlation between the presence of carbapenemase and the formation of biofilms was also evaluated.

### 4.7. DNA Extraction

The bacterial DNA was extracted from the isolated colonies using the boiling method: a bacterial inoculum was created in 500 µL of sterile water, vortexed for 15 s, incubated in a thermomixer for 10 min at 99 °C, and finally centrifuged at 12,000 rpm for 10 min. The supernatant containing the bacterial DNA was carefully collected without touching the pellet and stored at −21 °C for further use. The quality of the DNA was assessed from 1 µL extracted DNA by spectrophotometry, using the Eppendorf µCuvette^®^ G1.0 and D30 BioPhotometer (Eppendorf AG, Hamburg, Germany).

### 4.8. PCR for Detection of Carbapenemase Genes

For all strains included in the study, the presence of 7 carbapenemase genes and of one plasmid-mediated colistin resistance gene was evaluated by endpoint PCR, using the primers described in Table 3.

The presence of *bla*_KPC_, *bla*_OXA-48-like_, and *bla*_NDM_ was evaluated by triplex PCR and confirmed by simplex PCR. The PCR reaction mixes for the triplex and simplex PCR, as well as the amplification conditions, are presented Table 4.

The positive controls consisted of *K. pneumoniae* ATCC BAA 2470 (for *bla*_NDM_, 438 bp) and *K. pneumoniae* ATCC BAA 1705 (for *bla*_KPC_, 893 bp); well-characterized strains from the collection of the Microbiology department of the University of Medicine, Pharmacy, Science and Technology of Targu Mures, Romania were used for *bla*_OXA48-like_ (371 bp), *bla*_IMP_ (232 bp), and *bla*_VIM_ (390 bp).

The amplification products were visualized by electrophoresis in 1% agarose gel (Electran^®^DNA Agarose) containing GelRed^®^ (Biotium Inc., Fremont, CA, USA) for staining. Electrophoresis was performed in 1X TAE Buffer, for 45 min, at 100 V. The results were captured using MiniBIS Pro (DNR Bio-Imaging Systems Ltd., Jerusalem, Israel). The electrophoresis bands as well as the positive control bands were compared to the GeneRuler 100 bp (Thermo Fisher Scientific, Waltham, MA, USA) molecular ladder.

### 4.9. Statistical Analysis

The data were centralized and analyzed using Microsoft Excel 2021 and GraphPad Instat 3. A two-way ANOVA test was used to compare pairwise data, using multiple post hoc tests as follows: the post hoc Tuckey test was used to compare individual pairs, Bonferroni’s test was used for the comparison of the samples at 24 and 48 h, and Dunnett’s test was used for the comparison of samples to a control strain. The *t*-test was used to compare categorical data. For the correlation between the presence of carbapenemases and the formation of biofilms, Fisher’s test was used to calculate the relative risk and odds ratio.

## 5. Conclusions

This study highlights the significant variability in biofilm formation among carbapenem-resistant (CR) clinical isolates, particularly between screening and pathological samples. The results emphasize that clinical isolates tend to produce more intense biofilms compared to ATCC strains, potentially due to the genetic variability and adaptation following the colonization of the human body. Moreover, the higher prevalence of biofilm formation in clinical isolates compared to screening isolates suggests that biofilm production may play a crucial role in the infection severity and persistence. The presence of serum generally reduced biofilm formation, except for *Acinetobacter baumannii* and *Pseudomonas aeruginosa*, which exhibited increased biofilm production, and for some individual strains of other species. The carbapenemase genes were present in some of the clinical isolates but were not the main determinants of the general carbapenem resistance and were not associated with the bacterial ability to produce biofilms. All of these findings highlight the uniqueness of each bacterial isolate, emphasizing that it is not sufficient to identify the pathogen by the genus and species name, as the behavior is characteristic of each individual strain. These findings have important implications for treatment and infection control strategies, particularly in critical care settings, where CR infections are more prevalent.

## Figures and Tables

**Figure 1 antibiotics-13-00687-f001:**
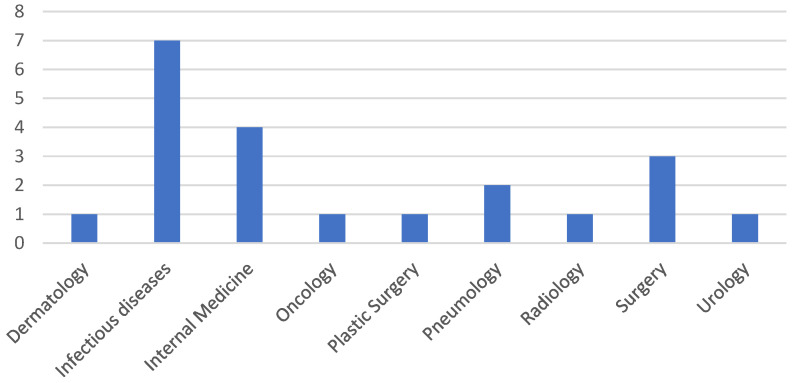
The frequency of screening isolates from different hospital wards.

**Figure 2 antibiotics-13-00687-f002:**
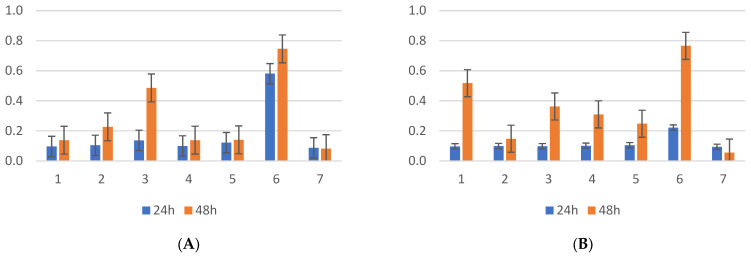
(**A**) *K. pneumoniae* screening strains incubated at 24–48 h without bovine serum; (**B**) *K. pneumoniae* screening strains incubated at 24–48 h with bovine serum; position 6—*K. pneumoniae* ATCC BAA 2470 (CPE-positive strain), position 7—*K. pneumoniae* ATCC 13883 (wild type).

**Figure 3 antibiotics-13-00687-f003:**
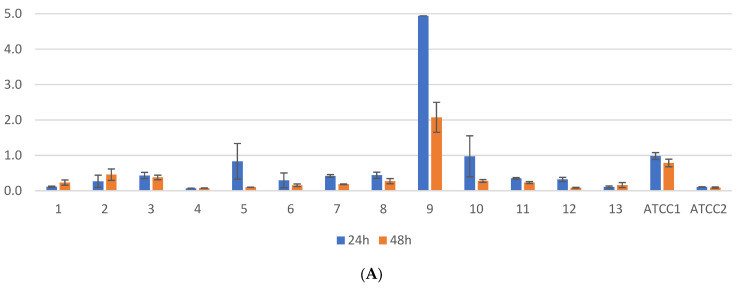
(**A**) *K. pneumoniae* clinical isolates incubated at 24–48 h without bovine serum; (**B**) *K. pneumoniae* clinical isolates incubated at 24–48 h with bovine serum; ATCC1—*K. pneumoniae* ATCC BAA 2470 (CPE-positive strain), position ATCC2—*K. pneumoniae* ATCC 13883 (wild type).

**Figure 4 antibiotics-13-00687-f004:**
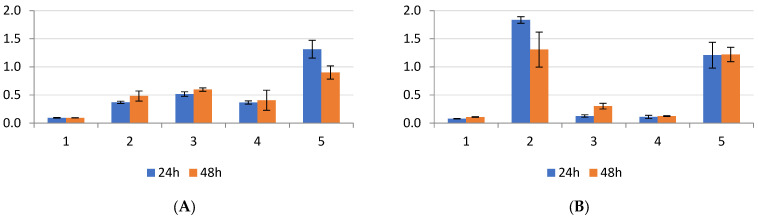
(**A**) *P. aeruginosa* screening strains incubated at 24–48 h without bovine serum; (**B**) *P. aeruginosa* screening strains incubated at 24–48 h with bovine serum; position 5—*P. aeruginosa* ATCC 27853.

**Figure 5 antibiotics-13-00687-f005:**
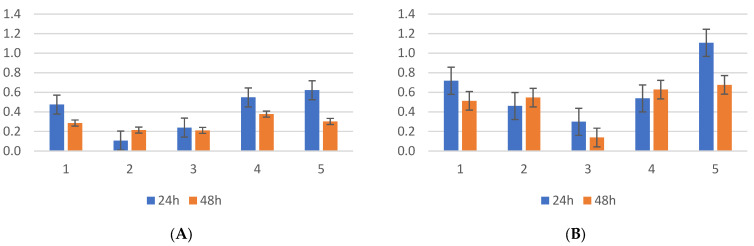
(**A**) *P. aeruginosa* clinical isolates incubated at 24–48 h without bovine serum; (**B**) *P. aeruginosa* clinical isolates incubated at 24–48 h with bovine serum; position 5—*P. aeruginosa* ATCC 27853.

**Figure 6 antibiotics-13-00687-f006:**
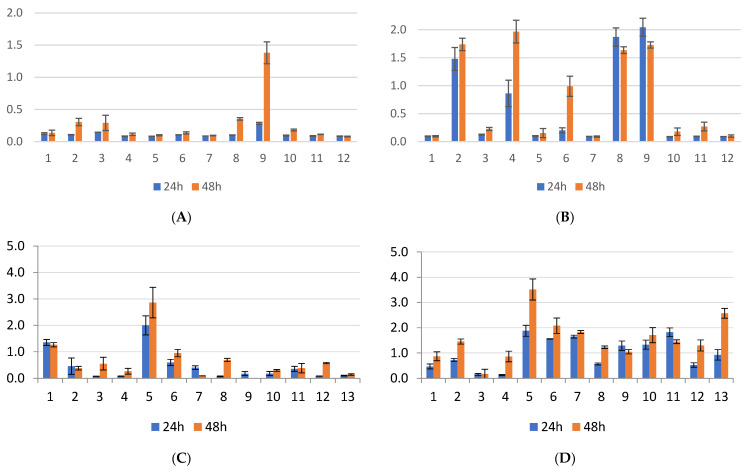
(**A**) *A. baumannii* screening isolates incubated at 24–48 h without bovine serum; (**B**) *A. baumannii* screening isolates incubated at 24–48 h with bovine serum; position 12 (**A**,**B**)—NTC; (**C**) *A. baumannii* clinical isolates incubated at 24–48 h without bovine serum; (**D**) *A. baumannii* clinical isolates incubated at 24–48 h with bovine serum.

**Figure 7 antibiotics-13-00687-f007:**
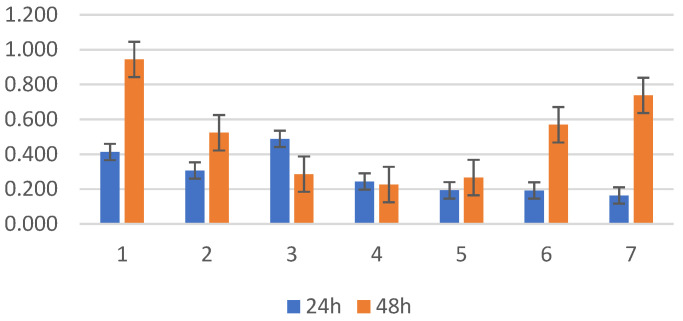
(**A**) *S. maltophilia* screening isolates incubated at 24–48 h without bovine serum; (**B**) *S. maltophilia* screening isolates incubated at 24–48 h with bovine serum; position 7—*S. maltophilia* ATCC 17666.

**Table 1 antibiotics-13-00687-t001:** Cumulative antibiogram for the tested isolates, showing their resistance percentages.

% Resistance	IMP	MEM	CN	AK	CIP	LEV	SXT	TZP*	CZA*	IMR*	CS	Other Beta-Lactams
Screening Non-fermentative	100%	100%	90%	93%	100%	100%	100%	100%	100%	-	-	-
Infection Non-fermentative	100%	100%	100%	100%	100%	100%	100%	100%	100%	100%	9%	-
Screening *K. pneumoniae*	100%	100%	100%	75%	100%	100%	100%	-	-	-	-	100%
Infection *K. pneumoniae*	100%	100%	100%	100%	100%	100%	100%	100%	100%	100%	62%	100%

Screening—CR isolates cultured from pharyngeal or rectal swabs. Infection—CR isolates cultured from pathological samples. IMP—Imipenem; MEM—Meropenem; CN—Gentamycin; AK—Amikacin; CIP—Ciprofloxacin; LEV—Levofloxacin; SXT—Trimethoprim–Sulfamethoxazole; TZP*—Piperacillin–Tazobactam; CZA*—Ceftazidime–Avibactam; IMR*—Imipenem–Relebactam; CS—Colistin; Other Beta-Lactams—includes Ampicillin, Amoxicillin–Clavulanic Acid, Cefuroxime, Ceftazidime, Cefotaxime, Ceftriaxone, Cefpodoxime, Cefepime, Ceftaroline. * for *P. aeruginosa* only.

**Table 2 antibiotics-13-00687-t002:** The number of isolates from screening and clinical samples.

Strain	Sample Type	Number of Strains	*bla* _OXA-48-like_	*bla* _KPC_	*bla* _NDM_	*bla* _IMP_	*bla* _VIM_	*bla* _SPM_
*K. pneumoniae*	Screening	*n* = 5	0	0	0	0	1	0
	Infection	*n* = 13	10	0	4	0	0	0
*A. baumannii*	Screening	*n* = 11	0	0	0	0	0	0
	Infection	*n* = 13	1	0	0	0	0	0
*P. aeruginosa*	Screening	*n* = 4	0	0	0	0	0	0
	Infection	*n* = 4	1	0	0	0	0	0
*S. maltophilia*	Screening	*n* = 6	0	0	0	0	0	0
	Infection	*n* = 0	-	-	-	-	-	-

Screening—CR isolates cultured from pharyngeal or rectal swabs. Infection—CR isolates cultured from pathological samples.

**Table 3 antibiotics-13-00687-t003:** The primer sequences [43].

	Gene Type	Primer Sequence (5′ > 3′)	Amplicon Length (bp)
Triplex PCR	*bla*_KPC_ Forward *bla*_KPC_ Reverse	ATGTCACTGTATCGCCGTCT TTTTCAGAGCCTTACTGCCC	893
	*bla*_OXA48-like_ Forward *bla*_OXA48-like_ Reverse	GCGTGGTTAAGGATGAACAC CATCAAGTTCAACCCAACCG	438
	*bla*_NDM_ Forward *bla*_NDM_ Reverse	GGTTTGGCGATCTGGTTTTC CGGAATGGCTCATCACGATC	621
Simplex PCR	*bla*_IMP_ Forward *bla*_IMP_ Reverse	GGAATAGAGTGGCTTAAYTCTC GGTTTAAYAAAACAACCACC	232
	*bla*_VIM_ Forward *bla*_VIM_ Reverse	GATGGTGTTTGGTCGCATA CGAATGCGCAGCACCAG	390
	*bla*_SPM_ Forward *bla*_SPM_ Reverse	AAAATCTGGGTACGCAAACG ACATTATCCGCTGGAACAGG	271
	*mcr*-1 Forward *mcr*-1 Reverse	CGGTCAGTCCGTTTGTTC CTTGGTCGGTCTGTAGGG	309

**Table 4 antibiotics-13-00687-t004:** The amplification conditions for carbapenemase genes.

	Initial Denaturation	Denaturation	Annealing	Elongation	Final Elongation
	1 Cycle	36 Cycles			1 Cycle
SPM, IMP	95 °C	95 °C	52 °C	72 °C	72 °C
	5 min	30 s	60 s	2 min	8 min
VIM	95 °C	95 °C	59 °C	72 °C	72 °C
	15 min	30 s	90 s	90 s	10 min
KPC, OXA-48-LIKE, NDM	95 °C	95 °C	57.3 °C	72 °C	72 °C
	10 min	30 s	40 s	50 s	5 min

## Data Availability

Data are contained within the article.

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
