# Peer review of "Biofilm Formation and Antibiotic Resistance Profiles in Carbapenemase-Producing Gram-Negative Rods—A Comparative Analysis between Screening and Pathological Isolates"

_antibiotics, 2024, doi:10.3390/antibiotics13080687_

Round 1

Reviewer 1 Report

Comments and Suggestions for Authors

I revised the manuscript. The manuscript introduce voluale information to literature. But manuscript must be revised extensively for Enflish grammer.

I suggested some correctios on the manuscript.

Comments on the Quality of English Language

Author Response

Thank you for your kind suggestion. We have managed to make the modifications, point by point as you recommended

I revised the manuscript. The manuscript introduces valuable information to literature. But manuscript must be revised extensively for English grammar. I suggested some correctios on the manuscript.

Answer: Thank you for the kind suggestions, we have included them in the manuscript.

Reviewer 2 Report

Comments and Suggestions for Authors

The manuscript is relevant for the field and scientifically sound but although the experimental design is appropriate, the details given in the methods section are incomplete and deficient.

The figures and tables are appropriate, and properly show specific data; I would recommend reducing the number of figures.

The manuscript’s results are reproducible based on the details given in the methods section, however some details regarding the results are missing.

The conclusions are consistent with the evidence and arguments presented appropriately.

The cited references are mostly recent publications and relevant.

Ethics statements are adequate.

Appendix A and Supplementary material are missing.

Extensive editing of English language required.

Please refer to the comments in the attached pdf file.

Comments on the Quality of English Language

Extensive editing of English language required.

Author Response

Thank you for your kind suggestion. We have managed to make the modifications, point by point as you recommended

The manuscript is relevant for the field and scientifically sound but although the experimental design is appropriate, the details given in the methods section are incomplete and deficient.

Answer: Thank you, we have improved the manuscript following your suggestions, and we are responding to the described issues point by point

The figures and tables are appropriate, and properly show specific data; I would recommend reducing the number of figures.

Answer: Thank you for your suggestion. If possible, we would like to maintain the figures because it is easier to follow the results described in the text by referring to the figures. Moreover, another reviewer suggested to add gel electrophoresis images, which were added in the appendix file.

The manuscript’s results are reproducible based on the details given in the methods section, however some details regarding the results are missing.

Answer: We have revised the results and added data on the antibiotic susceptibility, which was missing.

The conclusions are consistent with the evidence and arguments presented appropriately.

The cited references are mostly recent publications and relevant.

Ethics statements are adequate.

Answer: Thank you for the kind feedback

Appendix A and Supplementary material are missing.

Answer: Appendix A and supplementary material is included together with the manuscript.

Extensive editing of English language required.

Answer: English spellcheck, and grammar was improved by a native speaker.

Please refer to the comments in the attached pdf file.

Answer: Thank you for the comments, we have addressed them point by point.

Reviewer 3 Report

Comments and Suggestions for Authors

Comments on the Quality of English Language

Author Response

Thank you for your kind suggestion. We have managed to make the modifications, point by point as you recommended

The manuscript by Vintila et al adds to the knowledge on the presence of MDR bacterial strains prevalent in clinical setting and their biofilm formation capacity. This is a topic of worldwide concern and hence this type of studies are important. The authors employ the conventional approach for deification of MDR strains and used the end point PCR for detection of carbapenem resistance genes.  The manuscript still needs significant improvements. My comments regarding the same are as follows:

Abstract: Italicize all the bacterial species names and the antibiotic resistance genes.

Answer: Thank you, we have reviewed all the occurrences

Line 55: check the preposition

Answer: Thank you

Lines 58-61: not clear what the authors means. rewrite it

Answer: Thank you, we have rephrased as: Biofilms are complex communities of bacteria typically attached to a surface. They differ physiologically from commensal (or planktonic) bacteria, which have the potential to transition into biofilm-forming bacteria under certain conditions

Introduction: is not clear with several textual overlaps. Lack of fluency in writing makes it difficult and confusing for the reader.

Answer: Thank you for your kind advice, we modified some information for better fluency

line 102 he should be replaced by The

Answer: Thank you

In the biofilm formation section of results, it would be better if the authors performed pairwise comparisons of means for example a Tukeys test could have been applied.

Answer: Yes, we have used this test also, as presented for example here: „Post hoc Tukey-Kramer tests identified differences between the strains, with similar results at 48 hours”. Also, we have detailed the statistical analysis in section 4.9.

Table 1; describe screening and infection in the footnote

Answer: Thank you, we have detailed the screening and infection meaning in the footnote

Methods: line 325-326: what were the classical phenotypic methods used? Can the authors provide a reference to it.

Answer: Thank you, we have detailed the methodology

Lines 341-344: The results of antimicrobial susceptibility testing were not found in the present manuscript. The authors are advised to provide a table mentioning the level of phenotypic resistance of each strain.

Answer: Here is the corrected version of your sentence: Thank you. Indeed, there was a lack of information. We have introduced a table to present the phenotypic resistance and revised the results by adding data on the antibiotic susceptibility, which was missing.

Lines 382-383: The results of correlation between the presence of carbapenemase and formation of biofilms was not found.

Answer: Thank you, the information of the correlation was presented in the Genetical results. We have rephrased the sentence as: “There were 11 strains with genes that produced biofilm with OD above 0.1, with no association between the presence of carbapenemase gene and the ability of bacteria to produce biofilm (OR=1.4, p=1.0)” for better understanding

The authors are also advised to provide gel images of PCR products in supplementary

section.

Answer: Thank you, the images were added in the Appendix A

Please provide reference to biofilm growth assay method, acid acetic treatment, evaluation of carbapenems production, DNA extraction, PCR detection

Answer: Thank you, the references were added

Round 2

Reviewer 1 Report

Comments and Suggestions for Authors

I revised the manuscript based on previous comments, and The manuscript cannot require further revision.

Reviewer 2 Report

Comments and Suggestions for Authors

Dear all, the modifications are in line with the addressed comments. I would suggest accept the manuscript for publication.

Reviewer 3 Report

Comments and Suggestions for Authors

I have no further comments to make. The authors have improved the manuscript as per the recommendations

Comments on the Quality of English Language

Minor edits required.